# Non-Autoregressive Document-Level Machine Translation

**Guangsheng Bao**[1,2,*], **Zhiyang Teng**[3,*], **Hao Zhou**[4], **Jianhao Yan**[1,2], **and Yue Zhang**[2,5,†]

[1] Zhejiang University     [2] School of Engineering, Westlake University
[3] Nanyang Technological University     [4] Tsinghua University
[5] Institute of Advanced Technology, Westlake Institute for Advanced Study
[2] {baoguangsheng, yanjianhao, zhangyue}@westlake.edu.cn
[3] zhiyang.teng@ntu.edu.sg     [4] zhouhao@air.tsinghua.edu.cn

## Abstract

Non-autoregressive translation (NAT) models achieve comparable performance and superior speed compared to auto-regressive translation (AT) models in the context of sentence-level machine translation (MT). However, their abilities are unexplored in document-level MT, hindering their usage in real scenarios. In this paper, we conduct a comprehensive examination of typical NAT models in the context of document-level MT and further propose a simple but effective design of sentence alignment between source and target. Experiments show that NAT models achieve high acceleration on documents, and sentence alignment significantly enhances their performance.

However, current NAT models still have a significant performance gap compared to their AT counterparts. Further investigation reveals that NAT models suffer more from the multi-modality and misalignment issues in the context of document-level MT, and current NAT models struggle with exploiting document context and handling discourse phenomena. We delve into these challenges and provide our code at https://github.com/baoguangsheng/nat-on-doc.

## 1 Introduction

Non-autoregressive neural machine translation (NAT) models achieve significant acceleration on the inference, with even better translation performance, when compared to auto-regressive translation (AT) models on sentence-level MT (Gu et al., 2018, 2019; Stern et al., 2019; Ma et al., 2019; Lee et al., 2020; Huang et al., 2022; Shao and Feng, 2022). However, real applications (e.g., Google Translation and ChatGPT ) typically need to understand and respond in discourse, which requires a quick process of document-level content. Different from sentence-level MT, document-level MT considers a much larger inter-sentential context and models discourse dependence, such as anaphora, ellipsis, and lexical cohesion (Voita et al., 2019). The sentence-by-sentence translation done by NAT models cannot possibly produce contextual coherent translations, hindering the usage of NAT models in real scenarios.

Document-level context enhances the translation quality of AT models significantly (Werlen et al., 2018; Maruf et al., 2019; Liu et al., 2020; Bao et al., 2021; Sun et al., 2022; Bao et al., 2023). However, no precedent research explores the possibility of applying NAT models to document-level MT, which leaves open research questions including: 1) *Can NAT models leverage document context to improve translation quality?* 2) *Can NAT models generate translations with cross-sentential cohesion and coherence?* and 3) *Can NAT models achieve the same performance as AT models?* To address these questions, we investigate the challenges and opportunities of NAT models on document-level MT.

NAT models primarily face two challenges: **multi-modality** (Gu et al., 2018), which causes failure when one source has multiple possible translations, and **misalignment** (Saharia et al., 2020; Shao and Feng, 2022), which causes repetition and disorder in translations. Previous studies reduce modalities using knowledge distillation (Zhou et al., 2019) and improve alignment using new loss (Saharia et al., 2020; Shao and Feng, 2022; Huang et al., 2022). These methods have proven effective on the sentence level, while their efficacy is uncertain on the document level. First, document-level MT necessitates the modeling of cross-sentential coherence and discourse structure. Second, the extended length of input and output sequences intensifies the potential modalities. Last, the alignment between the source and target becomes harder because the enlarged input and output sequences increase the alignment space exponentially.

---

[*]Contributed equally.
[†]Corresponding author.

In this paper, we first assess recent NAT models on document-level MT tasks to understand their abilities and limitations, where we observe model failures and unstable performance. According to these observations, we further introduce **sentence alignment** to NAT models, which equips the encoder and decoder of the NAT models with the group-attention (Bao et al., 2021), restricting the target-to-source attention inside each target and source sentence pairs. Since each target sentence is aligned with a source sentence, we do not need to consider the possibility of aligning the target sentence with other source sentences. Therefore, the alignment space between the input and output sequences is significantly reduced.

Experiments on three benchmark datasets TED, News, and Europarl show that NAT models achieve a translation speed of more than 30 times faster than their AT counterparts on document-level MT, but still lag behind on the translation performance. Sentence alignment significantly enhances the translation performance for NAT models, closing the gap from 2.59 to 1.46 d-BLEU (document-level BLEU score) compared to the best AT results. To the best of our knowledge, we are the first to discuss the challenges and opportunities of NAT models in the context of document-level MT. We release our code and data to stimulate future research.

## 2 Related Work

**Non-Autoregressive Machine Translation.** Existing NAT models include three types: fully NAT models (Gu et al., 2018; Libovickỳ and Helcl, 2018; Qian et al., 2021; Huang et al., 2022), iterative NAT models (Ghazvininejad et al., 2019; Stern et al., 2019; Gu et al., 2019; Chan et al., 2020), and semi-autoregressive model (Ran et al., 2020). All these models generate tokens in parallel at some level. The fully NAT models generate tokens in a single forward pass, while the other two types generate tokens in multiple iterations or steps. In this paper, we take the fully NAT models to investigate document-level MT.

NAT models face two fundamental challenges: the multi-modality issue and the misalignment issue. The *multi-modality issue* happens when one source has multiple possible translations, breaking the conditional independence assumption. For example, "*thank you .*" in English can be translated into "*Vielen Dank .*" or "*Danke .*" in German. The second output token could either be "*Dank*" or ".",

which cannot be determined without a condition on the first token. The multi-modality issue causes vanilla NAT models to fail on complex datasets. Previous research leverages *knowledge distillation* to reduce modalities of a dataset, so that the conditional independence assumption is more likely satisfied (Gu et al., 2018). They generate a translation for each training sample using an AT model and then train NAT models on this generated training set. In this paper, we compare NAT models trained on both raw and knowledge-distilled data.

The *misalignment issue* elicits various NAT techniques. Vanilla NAT (Gu et al., 2018) uses an implicit token-alignment approach, enforcing a monotonic alignment between source and target by copy encoder outputs to decoder inputs. NAT+CTC (Libovickỳ and Helcl, 2018) introduces connectionist temporal classification (CTC) (Graves et al., 2006) to model the monotonic alignment explicitly. Others introduce non-monotonic alignment between source and target, such as n-gram matching (Shao et al., 2020; Shao and Feng, 2022), aligned cross-entropy (AXE) (Ghazvininejad et al., 2020), and order-agnostic cross-entropy (OAXE) (Du et al., 2021). In this paper, we investigate NAT models using the first two techniques.

**Document-Level Machine Translation.** Previous methods on document-level MT are dominated by auto-regressive models, which can be categorized into two approaches. The first approach splits a document into sentences and translates each sentence using its context as additional inputs (Zhang et al., 2018; Maruf et al., 2019; Zheng et al., 2021). The second approach takes a document as a whole translation unit and translates the document in one beam search (Liu et al., 2020; Bao et al., 2021; Sun et al., 2022; Bao et al., 2023). In this paper, we follow the second approach for our investigation of NAT models.

**Efficient Model for Long Sequence.** Recent advances in efficient models such as Longformer (Beltagy et al., 2020), Reformer (Kitaev et al., 2020), Linformer (Wang et al., 2020), and FlashAttention (Dao et al., 2022) improve the time and space complexity of the attention mechanism, which may affect the inference speed of both the auto-regressive and non-autoregressive MT models. In this paper, we focus on the relative inference acceleration of NAT compared to AT models with standard multi-head attention, leaving advanced attention mechanisms for the future.

## 3 Methods

### 3.1 AT Baselines for Document-Level MT

Document-level MT can be formulated as a seq2seq problem, where the input document $x$ and output document $y$ are represented in token sequence. The auto-regressive factorization is

$$p_\theta(y|x) = \prod_{t=1}^{T} p_\theta(y_t|y_{<t}, x), \qquad (1)$$

where $T$ denotes the number of target tokens and $y_t$ conditions on all previous tokens $y_{<t}$.

We choose two AT baselines.

**Transformer** (Vaswani et al., 2017) is the standard encoder-decoder model, which is widely used as a baseline. We represent each document as a sequence of tokens and train the model to do the seq2seq mapping.

**G-Transformer** (Bao et al., 2021) extends Transformer by separating the self-attention and cross-attention into sentence-level group-attention and document-level global-attention, as Figure 1 illustrates. The group-attention enables sentence alignment between input and output sequences.

**Knowledge Distillation (KD).** In addition to the raw data, we also experiment with KD data for training NAT models (Zhou et al., 2019), following previous NAT studies. We use G-Transformer fine-tuned on sentence Transformer as the teacher to generate distilled document translations.

### 3.2 Existing NAT Models

Due to the large amount of NAT models (a non-exhaustive search shows more than 60 recent NAT models proposed between 2020 and 2023), we follow the common practice (Huang et al., 2022; Qian et al., 2021) to choose representative NAT models for our investigation, leaving more complex iterative methods, semi-autoregressive methods, and pre-training settings for future. Specifically, we select representative fully NAT models for document-level experiments, where all the models are transformer-based and implemented in Fairseq (Ott et al., 2019).

**Vanilla NAT** (Gu et al., 2018) is the earliest NAT model, which factorizes translation into two parts

$$p_\theta(y|x) = p_\theta(T|x) \cdot \prod_{t=1}^{T} p_\theta(y_t|x), \qquad (2)$$

where the first part predicts the length $T$ of target $y$, the second part predicts target tokens $y_t$ given the length, and $x$ denotes the source input. For simplicity, we take the model as a whole and use the $\theta$ to denote all the parameters, which includes the parameters of the length model and the translation model.

An implicit token alignment is used between the source and target by copying the outputs of the encoder to the inputs of the decoder, where if the source and target have different lengths, they interpolate the decoder inputs uniformly. During training, it initializes the decoder with $T$ token positions, where $T$ denotes the real number of target tokens. During inference, it first predicts a number $T$ using $p_\theta(T|x)$, and then initializes the decoder with $T$ token positions to predict target tokens.

**GLAT** (Qian et al., 2021) improves vanilla NAT with a glancing mechanism, which adopts an adaptive glancing sampling strategy, exposing some target fragments to the decoder inputs so that the decoder can fit the training target easier. The glancing mechanism reduces the difficulty of independent training of target tokens and improves the performance on sentence-level MT significantly.

**Latent-GLAT** (Bao et al., 2022) further improves GLAT using latent variables, where the latent variables are supposed to alleviate the multi-modality problem. It represents the modality and alignment in the latent variables, dividing the learning of the target tokens into the modeling of the latent variables and the reconstruction of the target sequence.

**NAT+CTC** (Libovický and Helcl, 2018) applies connectionist temporal classification (CTC) (Graves et al., 2006) alignment to decoder outputs

$$p_\theta(y|x) = \sum_{a \in \beta(y)} \prod_{i=1}^{M} p_\theta(a_i|x), \qquad (3)$$

where $a$ denotes an alignment between $y$ and the reserved $M$ token positions in the decoder while $\beta(y)$ denotes all possible alignments. The alignment $a$ is assumed conditional independent given the source input $x$. The reserved token positions are usually set as times of the length of the source input.

CTC alignment in Eq. 3 requires a summation over all possible alignments, which is generally intractable. However, with conditional independence assumption for $a$, the summation can be achieved using dynamic programming.

**GLAT+CTC** (Qian et al., 2021) combines GLAT with CTC alignment.

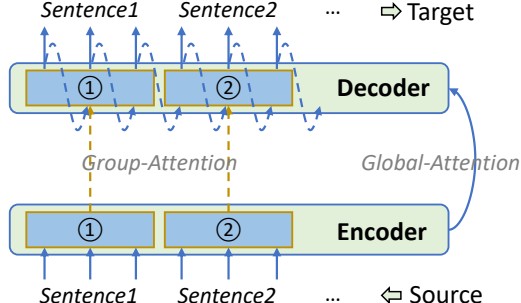

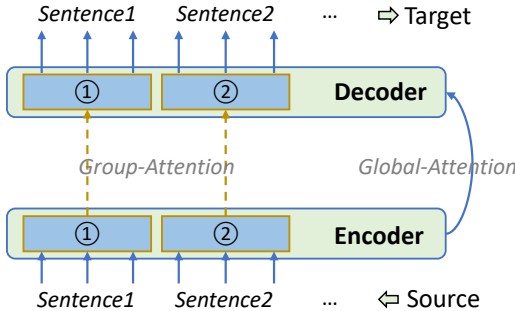

Figure 1: G-Transformer with sentence alignment between source and target documents, where the low layers use the group-attention and only the top 2 layers use the global-attention.

Figure 2: NAT model with sentence alignment, where we replace NAT encoder and decoder layers with G-Transformer layers and we remove the causal mask from G-Transformer decoder layers.

**DA-Transformer** (Huang et al., 2022) represents the output alignment in a directed acyclic graph (DAG), which contains vertices and edges. A path in DAG represents a possible alignment. It models translation as

$$p_\theta(y|x) = \sum_{a \in \beta(y)} p_\theta(a|x) p_\theta(y|a, x), \quad (4)$$

where $a$ denotes a path represented in a sequence of vertex indexes and $\beta(y)$ here denotes all paths having the same length as the target $y$. The right expression in Eq. 4 can further be expanded as

$$p_\theta(a|x) = \prod_{i=1}^{M-1} p_\theta(a_{i+1}|a_i, x), \quad (5)$$

$$p_\theta(y|a, x) = \prod_{i=1}^{M} p_\theta(y_i|a_i, x), \quad (6)$$

where $a_i$ are dependent on each other in a linear chain and $y_i$ are independent with each other given $a_i$ and $x$.

Similar to CTC alignment, DA-Transformer also adopts dynamic programming to marginalize all possible paths. The model provides three decoding strategies: greedy, lookahead, and beam search. In this paper, we choose lookahead decoding for balanced performance and inference speed.

### 3.3 NAT Models with Sentence Alignment

We propose novel NAT models with sentence alignment, as shown in Figure 2. The sentence alignment is a key feature in the G-Transformer (Bao et al., 2021), which is implemented through a group-attention module. The group-attention stabilizes self-attention and cross-attention in the model over long sequences. Inspired by the success, we

adopt it in our NAT models to reduce the alignment space.

Specifically, we adopt the G-Transformer encoder and decoder layers by removing the causal mask from self-attention in the decoder layers. We replace the encoder and decoder layers in NAT models with G-Transformer layers and redesign the initial output to include the special token of begin-of-sentence "" and end-of-sentence "" for each target sentence.

Substituting the causal masking layer in G-Transformer is only a necessary part at the attention level to facilitate sentence alignment. Its effectiveness hinges on both model design and training loss. Specifically, we introduce new length prediction per sentence in G-Trans+GLAT, improve CTC loss for sentence alignment in G-Trans+GLAT+CTC, and improve DAG design to restrict the transition in each sentence in G-Trans+DAT as follows.

**G-Trans+GLAT** predicts the target length per sentence instead of the whole sequence. It factorizes the translation as

$$p_\theta(y|x) = \prod_{j=1}^{K} \left[ p_\theta(T_j|x_j) \cdot \prod_{t=1}^{T_j} p_\theta(y_{j,t}|x_j, x) \right], \quad (7)$$

where $K$ denotes the number of sentences. We predict the length $T_j$ of the $j$-th target sentence and generate tokens $y_{j,t}$ accordingly. In addition to Eq. 2, where the corresponding source sentence of $y_t$ is unknown, $y_{j,t}$ also conditions on the source sentence $x_j$.

For each source sentence $x_j$, we predict the length $T_j$ of the target sentence using a linear classifier based on the mean pool of output features of the source sentence tokens. Consequently, we calculate the length $T$ of the target sequence by aggregating sentence lengths $T = \sum_{j=1}^{K} T_j$.

| Method | TED (d-BLEU) | | News (d-BLEU) | | Europarl (d-BLEU) | | Average (d-BLEU) | | Speed -up |
|---|---|---|---|---|---|---|---|---|---|
| | Raw | KD | Raw | KD | Raw | KD | Raw | KD | |
| *AT Baselines* | | | | | | | | | |
| G-Transformer (Bao et al., 2021) | **27.23** | **26.42** | **27.22** | **26.38** | **34.09** | **32.87** | **29.51** | **28.56** | - |
| Transformer (Vaswani et al., 2017) | 0.69 | 26.04 | 0.23 | 0.43 | 33.41 | 26.43 | 11.44 | 17.63 | 1.0x |
| *Existing NAT Models* | | | | | | | | | |
| Vanilla NAT (Gu et al., 2018) | 0.45 | 0.34 | 0.12 | 0.61 | 1.30 | 2.43 | 0.62 | 1.13 | **41.2x** |
| GLAT (Qian et al., 2021) | 1.77 | 0.03 | 0.01 | 2.56 | 2.13 | 8.17 | 1.30 | 3.59 | 40.0x |
| Latent-GLAT (Bao et al., 2022) | 2.10 | 1.87 | 0.30 | 2.30 | 5.26 | 4.25 | 2.55 | 2.81 | 29.2x |
| NAT+CTC (Libovickỳ and Helcl, 2018) | 21.54 | 24.98 | 15.95 | 24.03 | 0.00 | 31.58 | 12.50 | 26.86 | 27.7x |
| GLAT+CTC (Qian et al., 2021) | 18.49 | 25.31 | 10.23 | 20.01 | 0.00 | 29.47 | 9.57 | 24.93 | 26.3x |
| DA-Transformer (Huang et al., 2022) | 20.47 | 25.02 | 13.99 | 23.37 | 30.34 | 32.14 | 21.60 | 26.84 | 30.8x |
| *NAT Models with Sentence Alignment* | | | | | | | | | |
| G-Trans+GLAT (ours) | 19.96* | 24.23* | 15.14* | 23.04* | 25.67* | 31.45* | 20.26 | 26.24 | 30.0x |
| G-Trans+GLAT+CTC (ours) | **24.09*** | **26.31*** | **21.68*** | **25.61*** | 30.35* | 32.24* | **25.37** | **28.05** | 20.0x |
| G-Trans+DA-Trans (ours) | 23.45* | 25.73* | 21.43* | 24.70* | **30.36** | **32.29** | 25.08 | 27.57 | 25.1x |

Table 1: Main results on raw data (Raw) and knowledge distilled data (KD), where the failures are marked and investigated in section 5.1. We use the official code from the respective papers of NAT models, except NAT+CTC, which is implemented by ourselves based on the GLAT code. "*" denotes a statistical significance at the level of $p < 0.01$ using a t-test, compared to the corresponding baseline NAT model without sentence alignment.

**G-Trans+GLAT+CTC** integrates the sentence alignment with the CTC alignment. The default CTC algorithm aggregates all possible latent alignments across the entire sequence, which may align a source sentence to a wrong target sentence (e.g., the first source sentence to the second target sentence). Such global alignment not only slows down the training process but also causes unstable model performance. Different from the global alignment in Eq. 3, we apply CTC alignment on each sentence

$$p_\theta(y|x) = \prod_{j=1}^{K} \left[ \sum_{a_j \in \beta(y_j)} \prod_{i=1}^{M_j} p_\theta(a_{j,i}|x_j, x) \right], \quad (8)$$

where $M_j$ denotes the reserved token positions for the $j$-th target sentence. Since CTC alignment is restricted inside each target sentence, the alignment space $\beta(y_j)$ is enormously reduced compared to the $\beta(y)$ in Eq. 3.

**G-Trans+DA-Trans** introduces the sentence alignment into the output DAG of the DA-Transformer. The default DAG models the whole sequence, which enables the transition from a vertex in one sentence to a vertex in another sentence, making the transition space linearly increase with the enlargement of the sequence length.

To address the issue, we enforce a constraint to isolate the vertex transitions of each sentence, forcing the path $a$ on each sentence starting with a special vertex "" and ending with a special vertex "". The transition between sentences only happens from the vertex "" of a previous sentence to the vertex "" of the current sentence.

Formally, we use the same factorization as Eq. 4 but with a different collection $\beta(y)$ of paths. At the implementation level, we simply mask the transition matrix to disable transitions from one sentence to another sentence, so that the dynamic programming algorithm keeps unchanged.

## 4 Experiments

### 4.1 Experimental Settings

We evaluate the models using document-level MT benchmark (Maruf et al., 2019), which includes three datasets TED, News, and Europarl, representing three domains and various data scales for English-German translation. More details about each dataset and the preprocessing are in Appendix A.1. We follow Liu et al. (2020), evaluating the model performance in sentence-level BLEU score (*s-BLEU*) and document-level BLEU score (*d-BLEU*), which are explained in detail as Appendix A.2. We experiment on *Base* model (where Big model does not provide a stronger baseline) and evaluate the speedup using 1 GPU and 4 CPUs of a Tesla V100 environment as Appendix A.3.

### 4.2 Overall Results

As shown in Table 1, NAT models achieve high speedup but still suffer from a significant performance gap with their AT counterparts.

**Speedup.** The inference accelerations of NAT models on the document level are between 25x and 41x, which surpasses the accelerations on the sentence level (between 2x to 15x) by a big margin. Specifically, NAT and GLAT provide the biggest

acceleration by around 40x. More complex models, such as Latent-GLAT, DA-Transformer, and G-Trans+GLAT, accelerate inference by approximately 30x. CTC-based models have lower accelerations between 20x and 27x. On average, the acceleration on the document level is about 30x, which means that document-level MT systems could potentially save 96% computational resources and energy consumption by using NAT models.

**Performance.** Though NAT models underperform G-Transformer, some outpace the Transformer. The lengthy text sequence challenges both the AT models and the NAT models, resulting in exceptionally low d-BLEU scores (<10) in some settings. We treat these low scores as model failures and investigate them in section 5.1.

With the help of sentence alignment, the performance gap between NAT models and AT baselines is largely reduced, especially when trained on KD data. For example, G-Trans+GLAT+CTC achieves an average d-BLEU of 28.05 on KD data, which is only 0.51 points lower than the d-BLEU of 28.56 of G-Transformer. However, its performance on Raw data is 4.14 points lower than G-Transformer, suggesting that NAT models experience severe challenges with raw data. These results demonstrate that even though knowledge distillation and sentence alignment enhance NAT models largely, there is still a gap between the NAT models and the strongest AT baseline.

### 4.3 Breakdown Results

**Sentence-level alignment.** Sentence alignment substantially enhances the performance of NAT models. Specifically, for GLAT, sentence alignment enhances the performance on KD data from an average of 3.59 to 26.24 d-BLEU of G-Trans+GLAT. For GLAT+CTC, sentence alignment elevates the performance on KD data from 24.93 to 28.05 d-BLEU on average. This score is comparable to G-Transformer on KD data, leaving a minor gap of 0.51 d-BLEU on average. The better results of G-Trans+GLAT+CTC than G-Trans+GLAT suggest that sentence alignment complements output alignment techniques such as CTC.

The sentence alignment brings more benefits to models trained on raw data. As the average scores in Table 1 show, G-Trans+GLAT+CTC outperforms GLAT+CTC by 15.80 points on raw data and 3.12 points on KD data. G-Trans+DA-Trans outperforms DA-Transformer by 3.48 points on raw data and 0.73 points on KD data. These results suggest that sentence alignment mitigates the challenge brought by more modalities in raw data.

**Token-level alignment.** NAT models with implicit alignment such as vanilla NAT, GLAT, and Latent-GLAT fail on document-level MT, resulting in messy translations (repetitions, wrong words, and meaningless phrases). Conversely, NAT models with explicit alignments, such as NAT+CTC, GLAT+CTC, and DA-Transformer, produce superior document-level performance. DA-Transformer delivers the best overall performance among existing NAT models, even outperforming Transformer on Europarl (KD). These results suggest that token-level alignment plays an important role in NAT models to achieve good performance.

However, we also observe that training with CTC loss on documents is not always stable, leading to occasional failure and a high variance in their performance (e.g., +/-2.66 d-BLEU on raw data and +/-0.84 on KD data for GLAT+CTC). The failure happens more frequently on raw data than on KD data. We speculate that this instability is caused by the increased alignment space on long sequences, which can be mitigated by sentence alignment. Experiments on G-Trans+GLAT+CTC show stable training and lower variance in performance (+/-0.19 d-BLEU). These results suggest that token-level alignment alone is not enough to achieve stable training and good performance.

**Knowledge distillation.** Intuitively, documents have more modalities because they are much longer than sentences. If it is the case, knowledge distillation will become more critical for training NAT models on document-level MT than on sentence-level MT. We compare NAT models trained with and without knowledge distillation.

As Table 1 shows, NAT models trained on KD data generally outperform the same model trained on raw data. The improvement is especially significant for NAT models with explicit alignment. For example, DA-Transformer obtains an average of 26.84 d-BLEU on KD data, which is 5.24 points higher than 21.60 d-BLEU on raw data. In contrast, DA-Transformer on sentence-level MT achieves similar results on raw data and on KD data (Huang et al., 2022). The results suggest that compared to sentences, *documents have more severe multi-modality issues*.

Although knowledge distillation enhances the performance of NAT models, it also sets a ceiling

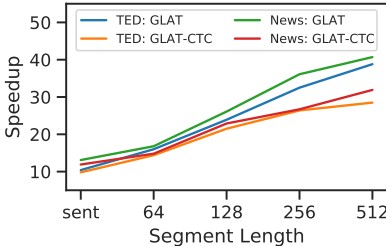

Figure 3: Speedup rises as long as the sequence length grows, evaluated on *TED* and *News*.

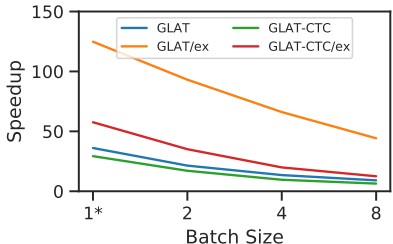

Figure 4: Speedup with various batch sizes evaluated on *News*, where the "/ex" conducts a strict calculation of speedup by excluding the time cost for model initialization from both the AT baseline and the NAT models.

to the performance on document-level MT. Comparing the performance of G-Transformer on KD data to raw data, we see that KD data downgrades the performance by about 1 d-BLEU on average (from 29.51 to 28.56). These results differ from previous results on sentence-level MT (Huang et al., 2022), where knowledge distillation maintains or even enhances the performance of AT models. The *discrepancy* calls for further research on techniques to reduce the modalities on the document level.

**Speedup on different sequence lengths.** We evaluate GLAT and GLAT+CTC on various sequence lengths, including the single sentence and segments with a maximum length of 64, 128, 256, and 512. As Figure 3 shows, the speedup displays consistent trends among different datasets and NAT models. The GLAT generally has a higher speedup than GLAT-CTC, especially when the segment length goes above 256. The speedups on TED and News are almost identical, which is expected because the time costs are supposed to be irrelevant to the data domains. The trends suggest that we benefit more from the inference acceleration on longer documents, where document-level MT tasks provide NAT models with the best scenario.

**Speedup on different batch sizes.** The previous study (Helcl et al., 2022) reports that NAT models have limited acceleration on the sentence-level MT in parallel inference settings. We evaluate the models in the settings with different batch sizes, including 1, 2, 4, and 8 instances per batch. Given that document-level MT sequences are much longer than sentence-level MT sequences, we do not evaluate using a batch size larger than 8.

As Figure 4 shows, when we increase the batch size from 1 to 8, the overall speedup of GLAT decreases from 40x to 9x. However, if we consider a more strict calculation of the speedup, excluding the time for initializing the models (which takes almost constant time) from the total evaluation time,

the inference speedup of GLAT is 125x and 44x for the batch size of 1 and 8, respectively. These results suggest that although the speedup ratio decreases for bigger batch sizes, NAT models on document-level MT still show significant acceleration in the parallel inference settings.

## 5 The Challenge of Long Sequence

The long input and output in document-level MT bring unexplored challenges to NAT models. In section 4, we learn that NAT models have a significant performance gap with their AT counterparts, and various NAT models even fail in some settings. We investigate these challenges in this section, leaving other discussions in Appendix D.

### 5.1 The Failure of NAT Models

Previous study (Bao et al., 2021) suggests that Transformer fails on long sequences because of local minima of loss values during the training process. We first investigate this possibility for the cause of the failure on NAT models. We evaluate GLAT, GLAT+CTC, Transformer, and G-Transformer on different lengths of sequences, as shown in Figure 5a. Overall, the four models produce different patterns of trends. The d-BLEU score of the Transformer rises as long as the sequence length increases until the failure happens at the length of 512 tokens. In contrast, the d-BLEU scores of the GLAT and GLAT+CTC descend progressively as long as the sequence length increases, which suggests *a performance decline instead of a training failure on NAT models*.

Further evidence on the bigger dataset Europarl *confirms* that the low performance of NAT models is different from the failure of Transformer. Bao et al. (2021) suggests that the local minima encountered by Transformer can be escaped by using a bigger dataset, resulting in normal scores on Europarl.

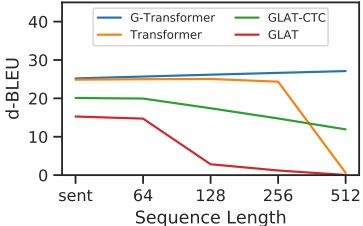
(a) *NAT and AT performance* on different sequence length.

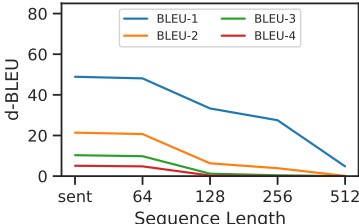
(b) *GLAT detailed BLEU scores* on different sequence lengths.

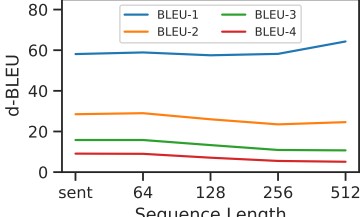
(c) *GLAT+CTC detailed BLEU scores* on different sequence lengths.

Figure 5: NAT vs. AT models on trends evaluated on News, where Transformer fails abruptly at the length of 512 while NAT models decrease progressively.

We evaluate Transformer, GLAT, and GLAT+CTC on Europarl, obtaining d-BLEU scores of 33.41, 2.13, and 0.00, respectively. We could see that GLAT and GLAT+CTC still give low scores on the bigger dataset, suggesting *a different cause from the local minima*.

We look into the d-BLEU score details, where BLEU-$n$ ($n \in [1, 2, 3, 4]$) denotes the precision on $n$-gram. As the d-BLEU scores on GLAT in Figure 5b shows, the BLEU-3/4 decreases rapidly, reaching almost 0.0 at the lengths of 256 and 512. The BLEU-1 remains relatively high at 27.3 on the length of 256, resulting in messy translations, where the generated tokens are related to the content but are repeated, disordered, or in the wrong collocation as shown in Table 2. In comparison, the d-BLEU scores on GLAT+CTC in Figure 5c decrease slowly, where the BLEU-1 even increases on the length of 512. We speculate the rapid decrease in BLEU-3/4 on GLAT is caused by the *multi-modality* and *misalignment* issues, which can be mitigated by explicit alignments, such as CTC.

**The exceptional zero scores.** Table 1 reveals some unexpected results. Both NAT+CTC and GLAT+CTC score a 0.00 d-BLEU on the raw Europarl dataset, a performance notably inferior to that of NAT and GLAT. This unprecedented anomaly may stem from the challenges of applying CTC loss on long sequences. The table further indicates that both combinations, NAT/GLAT+CTC and NAT/GLAT, experience training failures on the raw Europarl dataset. These failures are likely due to multi-modality and misalignment issues. These results empirically demonstrate that, while CTC loss can be effective, its application to document-level training is not consistently stable. We hypothesize that this instability arises from the expanded alignment space between lengthy input and output sequences, as detailed in the token-level alignment

**Source:** companies with a high percentage of floating rate debt stand to lose the most, Goldman said. outside pure stock plays, consumers stand to benefit as well through the rising dollar.

**Target:** Unternehmen mit einem hohen Anteil an flexiblen Zinsen werden am meisten verlieren, sagte Goldman. außerhalb der reinen Aktienspiele werden Verbraucher ebenfalls durch den steigenden Dollar profitieren.

**GLAT:** Unternehmen mit einem hohen , freien freien freien erverlieren die die meisten meisten , , . . te te dazu kräften profitieren profitieren profitieren profitieren profitieren durch durch steigenden steigenden profitieren profitieren .

**GLAT+CTC:** Unternehmen mit hohen der Schulden , die verlieren . außerhalb spielt Verbraucher den profitieren .

Table 2: GLAT and GLAT+CTC produce good translation at the beginning but downgrade later with repetitions and missing translations.

discussion in Section 4.3.

## 5.2 Document Context

We compare AT and NAT models with document context to the Transformer baseline without document context (trained on sentence) in Appendix B. The results suggest that document context enhances AT model (G-Transformer) by 0.70 s-BLEU. However, NAT models with document context still underperform the Transformer baseline, indicating that *current NAT models have a limited ability to utilize document context*.

We further apply an ablation study on the document context to quantify its contribution. As Table 3 illustrates, the performance of G-Trans+GLAT+CTC without document context drops by approximately 0.28 s-BLEU on average over the three benchmarks. Specifically, the target-side context contributes merely 0.01, while the source-side context contributes 0.27. The context contributions in G-Trans+GLAT+CTC are less than that in G-Transformer (0.23 and 0.70 s-BLEU on

| Method (s-BLEU) | TED | News | Europarl | Drop |
|---|---|---|---|---|
| G-Transformer ◇ | **25.12** | **25.52** | **32.39** | - |
| - target-side context | 25.05 | 25.41 | 32.16 | -0.14 |
| - source-side context | 24.56 | 24.58 | 31.39 | -0.70 |
| G-Trans+GLAT+CTC | 24.16 | **24.09** | **30.57** | - |
| - target-side context | **24.31** | 24.00 | 30.47 | -0.01 |
| - source-side context | 23.96 | 24.00 | 30.02 | -0.27 |

Table 3: Impact of *document context*, evaluated in s-BLEU. ◇ - the scores are from the paper report, where the model is trained on the raw datasets. The NAT models are trained on the KD datasets.

| Method | deixis | el.infl. | el.VP | lexcoh |
|---|---|---|---|---|
| Transformer (sent) ♡ | 50.0 | 53.0 | 28.4 | 45.9 |
| G-Transformer ◇ | **87.1** | **82.4** | **79.8** | **58.6** |
| G-Trans+GLAT+CTC | 50.0 | 55.2 | 46.6 | 45.9 |
| G-Trans+DA-Trans | 56.7 | 33.0 | 21.0 | 45.2 |

Table 4: *Discourse phenomena*. el.infl. - ellipsis of inflection. el.VP - ellipsis of verb phrase. lexcoh - lexical cohesion. ♡ - Transformer baseline trained on sentences. ◇ - G-Transformer baseline trained on documents.

target and source contexts, respectively). The minor contribution from the target-side context is expected, given that NAT models predict target sentences independently. The relatively low contribution of source-side context indicates that *NAT models do not fully exploit the source-side contextual information*.

**Leak of source context information on adjacent sentences?** We observe serious repetitions in the translations generated by NAT models without sentence alignment, raising the concern that the information in a source sentence may leak into its adjacent sentences during translation. We measure the repetitions on NAT models with sentence alignment, where the sentence boundaries are assured by the model design. We find that these models do not generate obvious cross-sentential repetitions. For example, on the TED test set, G-Trans+GLAT generates translations with repetition ratios of 0.14 and 0.02 for 1 and 2 grams, respectively, which are almost identical to the ratios of the reference.

### 5.3 Discourse Phenomena

We assess the discourse ability of NAT models using a human-annotated test suite in English-Russian MT (Voita et al., 2019). We train both the AT baselines and NAT models using the 1.5M document (only 4 sentences) pairs. In contrast to previous work (Bao et al., 2021), we do not use the additional 6M sentence pairs for training for the purpose of highlighting their discourse capabilities.

As Table 4 shows, the NAT models significantly underperform G-Transformer across the four discourse phenomena. The performance of G-Trans+GLAT+CTC matches the Transformer baseline (without document context) on deixis and lexical cohesion (lexcoh), but excels on the ellipsis of inflection (el.infl.) and ellipsis of verb phrase (el.VP). G-Trans+DA-Trans achieve relatively higher deixis than G-Trans+GLAT+CTC be-

cause its DAG link models the target-side dependence somehow. These results suggest that *current NAT models have some discourse abilities but still struggle with handling discourse phenomena*.

## 6 Conclusion and Future Work

We investigated NAT models on document-level MT, revealing more severe affections of multimodality and misalignment issues on documents than on sentences. We proposed NAT models with sentence alignment, reducing the possible alignment space, and achieving the best results across three benchmarks. Our experiments show that NAT models significantly accelerate text generation in documents while their performance still lags behind their AT counterparts. Further analysis shows that fully NAT models underutilize document context, leading to loose discourse relations.

As the first research of NAT models on document-level MT, we hope this work could stimulate future research on *reducing the modalities, exploiting document contexts, and modeling discourse dependence*.

## Limitations

We do not enumerate all recent NAT models. For the purpose of our investigation, we only evaluate the pure (fully) NAT models, leaving other NAT models, such as semi-autoregressive and iterative NAT models, out of our scope.

## Acknowledgements

We would like to thank the anonymous reviewers for their valuable feedback. This work is funded by the China Strategic Scientific and Technological Innovation Cooperation Project (grant No. 2022YFE0204900) and the National Natural Science Foundation of China (grant NSFC No. 62161160339). Zhiyang Teng is partially supported by CAAI-Huawei MindSpore Open Fund (CAAIXSJLJJ-2021-046A).

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

| Dataset | Sentences train / dev / test | Documents train / dev / test | Segments train / dev / test | Sents per Segment train / dev / test | Tokens per Segment train / dev / test |
|---|---|---|---|---|---|
| TED | 0.21M / 9K / 2.3K | 1.7K / 92 / 22 | 11K / 483 / 123 | 18.3 / 18.5 / 18.3 | 436 / 428 / 429 |
| News | 0.24M / 2K / 3K | 6K / 80 / 154 | 18.5K / 172 / 263 | 12.8 / 12.6 / 11.3 | 380 / 355 / 321 |
| Europarl | 1.67M / 3.6K / 5.1K | 118K / 239 / 359 | 162K / 346 / 498 | 10.3 / 10.4 / 10.3 | 320 / 326 / 323 |

Table 5: English-German datasets for evaluation, where we split each document into segments with a maximum of 512 tokens.

| Method (s-BLEU) | TED Raw | TED KD | News Raw | News KD | Europarl Raw | Europarl KD | Average Raw | Average KD |
|---|---|---|---|---|---|---|---|---|
| **AT Baseline** | | | | | | | | |
| Transformer (sent baseline) | 24.63 | - | 24.97 | - | 31.34 | - | 26.98 | - |
| G-Transformer (doc baseline) | **25.12** | - | **25.52** | - | **32.39** | - | **27.68** | - |
| **NAT Models** | | | | | | | | |
| G-Trans+GLAT | 17.05 | 21.81 | 13.37 | 21.16 | 23.49 | 29.66 | 17.97 | 24.21 |
| G-Trans+GLAT+CTC | 21.87 | **24.16** | 20.18 | **24.09** | 28.80 | **30.57** | 23.62 | **26.27** |
| G-Trans+DA-Trans | 21.02 | 23.48 | 19.79 | 23.10 | 28.64 | 30.55 | 23.15 | 25.71 |

Table 6: Sentence-level performance with and without document context.

## A  Experimental Settings

### A.1  Datasets

We evaluate the models using document-level MT benchmark (Maruf et al., 2019), which includes three datasets covering three domains and various data scales for English-German translation.

**TED** is from IWSLT17, which is transcribed from TED talks that each talk forms a document. *tst2016-2017* is used to test the model, and the rest for development.

**News** is from News Commentary v11 for training set. For testing and development sets, it uses *newstest2016* and *newstest2015*, respectively.

**Europarl** is from Europarl v7, where the training, development, and testing sets are randomly split.

Table 5 shows the detailed statistics. We preprocess the documents by tokenizing and truecasing using MOSES (Koehn et al., 2007) and applying BPE with 30,000 merge operations. We follow Bao et al. (2021) to split each document into segments with a maximum length of 512 tokens.

### A.2  Evaluation Metrics

We follow Liu et al. (2020), evaluating the model performance in *s-BLEU* and *d-BLEU*.

**s-BLEU** is calculated on each pair of sentences, which are obtained using the sentence alignment between source and target documents.

**d-BLEU** is calculated on each pair of segments, taking the whole segment as a translation unit to compute the BLEU score.

We calculate the BLEU score (Papineni et al., 2002) using sacreBLEU on the detokenized cased

words.

### A.3  Model Configuration

All the experiments are run on the *Base* model, which has 6 layers, 8 heads, 512 embedding dimensions, and 2048 hidden dimensions. We train the models on 4 Tesla V100/A100 GPUs for both AT and NAT. By default, we use the Tesla V100, but in case out-of-memory happens, we switch to Tesla A100 to re-train the model. We do not change the code of existing NAT models, and we obtain the default training and testing arguments from their official code. We update the arguments max-source-positions and max-target-positions to fit the enlarged input and output sequences. We run all main experiments three times and report the median.

We assess the speedup on the test set using a batch size of 1 within a virtual environment equipped with 1 GPU and 4 CPUs of a Tesla V100.

## B  Contribution of Document Context

Previous studies demonstrate that document context can significantly enhance MT performance (Zhang et al., 2018; Zheng et al., 2021; Bao et al., 2021). We evaluate its contribution as shown in Table 6. Comparing Transformer on sentence-level MT and G-Transformer on document-level MT, we can see that document context enhances s-BLEU by 0.49, 0.55, and 1.05 on TED, News, and Europarl, respectively. However, the best NAT results produced by G-Trans+GLAT+CTC on KD data are still lower than the Transformer baseline by 0.47, 0.88, and 0.77 on TED, News, and Europarl,

| | |
|---|---|
| **Source:** companies with a high percentage of floating rate debt stand to lose the most, Goldman said. outside pure stock plays, consumers stand to benefit as well through the rising dollar. savers could see gains as well through higher yields at the, though experts differ on how quickly that will take hold. | |
| **Target:** Unternehmen mit einem hohen Anteil an flexiblen Zinsen werden am meisten verlieren, sagte Goldman. außerhalb der reinen Aktienspiele werden Verbraucher ebenfalls durch den steigenden Dollar profitieren. Sparer könnten Gewinne durch höhere Erträge sehen, auch wenn Experten unterschiedlicher Meinung sind, wie schnell das stattfinden wird. | |
| **GLAT:** Unternehmen mit einem hohen , freien freien freien erverlieren die die meisten meisten , , . . . te te dazu kräften profitieren profitieren profitieren profitieren profitieren durch durch steigenden steigenden profitieren profitieren . die könnten könnten Gewinne Erträge höhere höhere erzielen erzielen erzielen , obwohl sich sich sich sich , , schnell schnell diese diese Fuß wird . werden werden . . . . . . . . der die die die , , , , , , , , , , erzielen erzielen erzielen , , sich sich sich sich , , , , sich sich sich sich , , , , , die die die , , halten nehmen . | |
| **G-Trans+GLAT:** Unternehmen mit einem hohen Prozentsatz freier Zinsen werden am am meisten verlieren , sagte Goldman Goldman . außerhalb reinreinen Aktien würden die Verbraucher ebenso durch durch den steigenden Dollar profitieren . die Sparer könnten durch höhere Erträge Erträge Gewinne erzielen , auch sich die Experten darüber unterscheiden , wie schnell das das einnehmen wird . | |
| **GLAT+CTC:** Unternehmen mit hohen der Schulden , die verlieren . außerhalb spielt Verbraucher den profitieren . könnten Gewinne höhere Renditen , sich Experten sich , schnell sich . Verbraucher Aktien werden wird Aktien . | |
| **G-Trans+GLAT+CTC:** Unternehmen mit einem hohen Prozentsatz freier Zinsen würden am meisten verlieren , sagte Goldman . reine Aktienspielen würden die Verbraucher durch den steigenden Dollar . Sparer könnten Gewinne ebenso durch höhere Renditen sehen , auch sich die Experten darüber sind , wie schnell dies haben wird . | |

Table 7: Case study. GLAT and GLAT-CTC show good translation quality at the beginning but poor quality at the end of the document. G-Trans+GLAT and G-Trans+GLAT+CTC show more consistent translation quality for sentences at different positions.

respectively, not to mention the G-Transformer baseline. These results indicate that NAT models have a limited ability to utilize document context.

## C   Case Study.

As a case in Table 7 shows, even though the source is not a lengthy document, sentence alignment enhances the translation quality significantly. Specifically, on the one hand, G-Trans+GLAT and G-Trans+GLAT+CTC estimate the target length more accurately than GLAT and GLAT+CTC because of their fine-grained length prediction on each target sentence. On the other, G-Trans+GLAT and G-Trans+GLAT+CTC show consistent translation quality between the beginning and the end of the document, while GLAT and GLAT+CTC show good quality at the beginning and poor quality at the end. This case illustrates the challenge of long text sequences to NAT models that the translation quality degrades as long as the distance to the beginning of the document increases, and demonstrates the advantage of sentence alignment to stop the degradation.

## D   More Discussion

**Can a larger model handle longer input sequences?** A larger model does not necessarily solve the longer input sequence issue given the same amount of training data. For pre-trained language models, a larger model typically shows a stronger ability to handle long sequences because the increased parameters enable the model to capture more complex long-context dependencies. However, for a non-pretraining setting, a larger model does not necessarily show better performance on the current document-level MT benchmarks. The longer input makes the model more likely to overfit and a larger model will make it worse given the limited training corpus. Take the AT baseline G-Transformer as an example. When we increase the model size from Base to Big, the d-BLEU scores decline by 0.36, 1.41, and 0.10 on TED, News, and Europarl, respectively. When we further increase the model size to Large, the d-BLEU scores further decline by 16.53, 8.49, and 0.56 on TED, News, and Europarl, respectively.

**Can the model handle more complex sentence alignments, such as one source sentence divided into multiple target simple sentences?** Actually, the current model can handle the case. The sentence-level alignment between the source and target is achieved by the same group tag assigned to the source tokens and the target tokens. We can treat the multiple simple sentences as a whole translation unit, assigning them the same group tag to map them to a single complex sentence.