# OpenReview forum: "Non-Autoregressive Document-Level Machine Translation"
_EMNLP/2023/Conference — EMNLP 2023 Findings_

### Official Review · Reviewer_HkNb · 2023-07-24

**Soundness:** 4

**Excitement:**

4: Strong: This paper deepens the understanding of some phenomenon or lowers the barriers to an existing research direction.

**Missing References:**

- "BLEU: a Method for Automatic Evaluation of Machine Translation" by Kishore Papineni, Salim Roukos, Todd Ward, and Wei-Jing Zhu
- "fairseq: A Fast, Extensible Toolkit for Sequence Modeling" by Myle Ott, Sergey Edunov, Alexei Baevski, Angela Fan, Sam Gross, Nathan Ng, David Grangier, and Michael Auli

**Paper Topic And Main Contributions:**

The paper innovatively explores the capabilities of non-autoregressive translation models in the context of document-level machine translation, an area that has not been explored in depth to date. It presents an effective yet simple strategy for sentence alignment between source and target, and evaluates the effectiveness of the models against one benchmark with three datasets. The authors provide a comprehensive examination of standard non-autoregressive translation models in the field of document-level machine translation and uncover the difficulties these models face in managing discourse phenomena and utilizing document context. And they developed a unique sentence alignment strategy that reduces potential alignment space, providing improvements in the established benchmark. Furthermore, they show that while non-autoregressive translation models significantly speed up decoding in documents, they currently lag behind their autoregressive translation models in terms of overall performance. As a weakness, in addition to the limitations mentioned by the authors, experiments with only one language pair are not very convincing. It would be better to have experiments with other language pairs as well as larger datasets.



**Questions For The Authors:**

- In Equation 2, is the parameter set \theta of the length model the same as that of the translation model?
- Line 357, why not "big" model for a stronger baseline?

**Reasons To Accept:**

- The paper investigates the potential of non-autoregressive translation models in document-level machine translation, an area that has not yet been thoroughly explored.
- It introduces a simple but effective sentence alignment strategy that reduces the possible alignment space and offers improvements over the document-level machine translation benchmark.
- It conducts a thorough investigation of typical non-autoregressive translation models in document-level machine translation, which can provide valuable insights for future research in this field.

**Reasons To Reject:**

- The experiments were performed on only one language pair and the datasets used are relatively small.

**Reproducibility:**

3: Could reproduce the results with some difficulty. The settings of parameters are underspecified or subjectively determined; the training/evaluation data are not widely available.

**Reviewer Confidence:**

3: Pretty sure, but there's a chance I missed something. Although I have a good feel for this area in general, I did not carefully check the paper's details, e.g., the math, experimental design, or novelty.

**Typos Grammar Style And Presentation Improvements:**

- The abbreviation "KD" is first used at line 192, so it may be introduced there instead of in the caption of Table 1.
- Line 208, x may already be introduced with Equation 1.
- Line 259, does the \beta(y) define the same as that in Equation 3? If so, it may already be introduced with Equation 3.
- Line 319, K is already used in Equation 7 and may therefore be introduced there.
- Line 355, the term "d-BLEU" is already used in Section 1, which may be introduced there.
- Line 591, the abbreviation VP is not introduced.

---

> ### Author Rebuttal · Authors · 2023-08-29
>
> Thank you for your constructive feedback.
>
> **Q1:** Experiments on more language pairs and bigger datasets? \
> **A1:** We will revise the paper to include one more language pair and a bigger dataset. Specifically, we consider Zh-En literary translation from WMT23, which contains 1.9M sentences. Our preliminary experiments show positive results as for En-De datasets: failure happens on GLAT, CTC improves the performance, and sentence alignment further enhances the model. We will add a new section to talk about it in the revised version if the space allows.
>
> The latest document-level MT datasets available are at the same level as the Europarl v7 dataset (1.7M sentences, 118K documents) that we use. For example, Zh-En literary translation from WMT23 has about 1.9M sentences, and En-De Europarl v10 from WMT23 has about 1.8M sentences. A larger dataset could probably better demonstrate the advantage of our model on document-level MT, and we will discuss the potential impact.
>
>
> **Q2:** In Equation 2, is the parameter set \theta of the length model the same as that of the translation model? \
> **A2:** For simplicity, we take the model as a whole and use the \theta to denote all the parameters, which includes the parameters of the length model and the translation model. We will revise the paper to clarify it.
>
> **Q3:** Line 357, why not “big” model for a stronger baseline? \
> **A3:** A bigger model does not necessarily provide a stronger baseline on the current document-level MT benchmarks because a longer input makes the model more likely to overfit and a bigger model will make it worse given the limited training corpus. Take the AT baseline G-Transformer as an example. When we increase the model size from Base to Big, the d-BLEU scores decline by 0.36, 1.41, and 0.10 on TED, News, and Europarl, respectively. When we further increase the model size to Large, the d-BLEU scores further decline by 16.53, 8.49, and 0.56 on TED, News, and Europarl, respectively. We will add the discussion in the revised version.
>
> **Q4:** Missing references. \
> **A4:** We will add the missing references on BLEU and Fairseq.
>
> **Q5:** Typos and presentation. \
> **A5:** We will fix the issues and carefully proofread our paper again.

---

### Official Review · Reviewer_3tvA · 2023-08-03

**Soundness:** 3

**Excitement:**

3: Ambivalent: It has merits (e.g., it reports state-of-the-art results, the idea is nice), but there are key weaknesses (e.g., it describes incremental work), and it can significantly benefit from another round of revision. However, I won't object to accepting it if my co-reviewers champion it.

**Missing References:**

[1] https://aclanthology.org/2022.emnlp-main.466.pdf

[2] https://proceedings.neurips.cc/paper_files/paper/2022/file/67d57c32e20fd0a7a302cb81d36e40d5-Paper-Conference.pdf

[3] https://arxiv.org/pdf/2011.04006.pdf

[4] https://aclanthology.org/2022.naacl-main.129.pdf

[5] https://arxiv.org/pdf/2205.10577.pdf

**Paper Topic And Main Contributions:**

This paper develops two points: 1) an evaluation of current non-autoregressive methods on document-level MT 2) an proposed technique to use sentence alignments so the MT decoder works at sentence level while the encoder at document-level.
The selected NAT methods are implemented in the same backbone model (base model from FairSeq). One important missing comparison is: Xlm-d which showed higher scores than the studied ones although having a different backbone model. The sentence alignment shows consistent improvements with respect to the full document decoding. In all evaluations significance scores are missing.


**Questions For The Authors:**

[Q1]: One limitation of the proposed approach is assuming that there will be one-to-one alignment of sentences on target-and-source. This is true in current benchmark due to they were originally build  for sentence-level translation. However it's not true for all language pairs and writing styles. Some language allows for more complex sentences which in english will be divided into several simple sentences.  What future solution or modification in your solution do you see?
[Q2]: NAT models are not always fluent, even if the BLEU scores are high. As you point out in your analysis, with discourse metrics. Dis you measure any issues of repetitions cross-sentences, e.g. leak of source context information on adjacent sentences?

**Reasons To Accept:**

- This work establishes a benchmark for NAT at document-level.

- Consistent improvement shows by decoding sentences instead of documents.


**Reasons To Reject:**

- Significance scores are missing to make the work complete and to validate several conclusions in the results description about which method works better.

- One concern is that current research in document-level machine translation, including this work, does not update with recent advances on efficient methods for supporting long inputs [2,3]. This may have an impact on the reported speeding times and also training efficiency. In my view one main contribution of this paper is to serve as a benchmark for current methods but it’s not exhaustive with current advances.

- Another concert is that this work does not include perspectives on realistic scenarios and concerts for measurement speed for NAT like batch sizes and more efficient implementations [4,5]. And consistent evaluation metrics [5]. In my view one main contribution of this paper is to serve as a benchmark but it’s not exhaustive with current advances.


**Reproducibility:**

4: Could mostly reproduce the results, but there may be some variation because of sample variance or minor variations in their interpretation of the protocol or method.

**Reviewer Confidence:**

4: Quite sure. I tried to check the important points carefully. It's unlikely, though conceivable, that I missed something that should affect my ratings.

---

> ### Author Rebuttal · Authors · 2023-08-29
>
> Thank you for your constructive feedback.
>
> **Q1:** In my view, one main contribution of this paper is to serve as a benchmark for current methods, but it’s not exhaustive with current advances. \
> **A1:** We will revise the paper to clarify this. Due to the large amount of NAT models (a non-exhaustive search shows more than 60 recent NAT models proposed between 2020 and 2023), existing studies, including XLM-D (Wang et al., 2022), DA-Transformer (Huang et al., 2022), and GLAT (Qian et al., 2021), generally do not conduct an exhaustive comparison with previous methods but instead choose several representatives. We follow the practice and select representative pure (fully) NAT models (Line198) for our investigation -- **understanding the possibility and key challenges to applying NAT to document-level MT**. We leave more complex iterative methods,  semi-autoregressive methods, and pre-training settings for future work (Line616) since they involve external factors, which may complicate the analysis. We hope our preliminary work on pure NAT models can inspire further research on more complex NAT methods.
>
> **Q2:** Current research in document-level MT, including this work, does not update with recent advances in efficient methods for supporting long inputs. This may have an impact on the reported speeding times and training efficiency. \
> **A2:** We will revise the paper to discuss it. A major challenge for document-level MT is the large alignment space between the long input and output sequences, which may cause the training failure of AT and NAT models. Recent advances in efficient models such as Longformer, Reformer, Linformer, and FlashAttention mainly focus on improving the time and space complexity of the attention mechanism, which does not **address the alignment issue** in document-level MT. As the previous study (G-Transformer) shows that although the local attention mechanism influences the model performance by about 1.78 d-BLEU (average across TED, News, and Europarl), the sentence-level alignment dominates the performance by about 14.68 d-BLEU.
>
> Due to space limitation, we mainly discuss the relative inference acceleration of NAT compared to AT models. Therefore, we leave out general techniques such as Longformer, Reformer, Linformer, and FlashAttention, because these techniques can be applied to both AT and NAT models. We will add relevant discussion.
>
> **Q3:** This work does not include perspectives on realistic scenarios and concerts for measurement speed for NAT like batch sizes and more efficient implementations. \
> **A3:** We will add comparisons on different batch sizes. Our experiments show that when we increase the batch size from 1 to 4 and 8, the overall speedup of GLAT decreases from 40x to 16x and 11x. If we exclude the time for initializing the models (which takes almost constant time), the inference speedup of GLAT will be 124x, 66x, and 44x for the batch size of 1, 4, and 8, respectively.
>
> More efficient implementations such as FlashAttention can be applied to both AT and NAT models; therefore, we leave them out of our scope and focus on NAT-related techniques.
>
> **Q4:** Did you measure any issues of repetitions cross-sentences, e.g., leak of source context information on adjacent sentences? \
> **A4:** We will discuss it in the revised version. We observe serious repetitions on NAT models **without** sentence alignment but we cannot measure the cross-sentence repetitions reliably because these models do not always generate sentence boundaries. We measure the cross-sentence repetitions on NAT models **with** sentence alignment, where the sentence boundaries are assured by the model design. We find that they do not generate obvious cross-sentence repetitions. For example, on the TED test set, G-Trans+GLAT generates translations with cross-sentence repetition ratios of 0.14 and 0.02 for 1 and 2 grams, respectively. G-Trans+GLAT+CTC generates translations with cross-sentence repetition ratios of 0.15 and 0.02 for 1 and 2 grams, respectively. In comparison, the reference translations have cross-sentence repetition ratios of 0.15 and 0.02 for 1 and 2 grams, respectively, which are almost the same as G-Trans+GLAT and G-Trans+GLAT+CTC.
>
> **Q5:** Some language allows for more complex sentences which in English will be divided into several simple sentences. What future solution or modification in your solution do you see? \
> **A5:** We appreciate your insightful suggestion. Actually, the current model can handle the case. The sentence-level alignment between the source and target is achieved by the same group tag assigned to the source tokens and the target tokens. We can treat the multiple simple sentences as a whole translation unit, assigning them the same group tag to map them to a single complex sentence.
>
> **Q6:** Significance scores are missing. \
> **A6:** We will revise the paper to include significance scores. G-Trans+GLAT and G-Trans+GLAT+CTC outperform GLAT and GLAT+CTC on TED, News, and Europarl (both Raw and KD) with a statistical significance at the level of p<0.01 using t-test. G-Trans+DA-Trans outperforms DA-Trans on TED and News (Raw and KD) with a statistical significance at the level of p<0.01.
>
> **Q7:** Missing references. \
> **A7:** Thanks for the suggestions and we will add and discuss them. We will add the missing references. We will add XLM-D [1] as a representative iterative model in related work. We will add a new paragraph in related work to discuss efficient Transformers for long sequences, taking FlashAttention [2] and Long Range Arena [3] as two related studies to discuss their relation and difference to our study. We will add a new paragraph in Section 4.3 – Breakdown Results to discuss the speedup on different batch sizes, taking previous studies [4] [5] as references.

---

### Official Review · Reviewer_NAdw · 2023-08-14

**Soundness:** 4

**Ethical Concerns:**

Yes

**Excitement:**

4: Strong: This paper deepens the understanding of some phenomenon or lowers the barriers to an existing research direction.

**Paper Topic And Main Contributions:**

The primary contribution of this research is the examination of Neural Autoregressive Translation (NAT) in document-level machine translation tasks. This is accomplished by adapting the G-Transformer model from its causal decoder version to the NAT decoder version.

**Questions For The Authors:**

Do authors have some insight on how the model performances in terms of the model size? Does the larger model can handle the longer input sequence issue? Large model may be more powerful to deal with the multi-modal and misalignment issue. In other word, is this study only useful for small or middle size AT and NAT models?

Why NAT+CTC and GLAT+CTC is worse than NAT and GLAT for Europarl dataset in Table 1?

**Reasons To Accept:**

This paper undertakes a captivating exploration of the Neural Autoregressive Translation (NAT) model in the context of document-level machine translation tasks. The experimental results highlight that the proposed G-Transformer-based NAT model surpasses the performance of the Transformer model across three benchmark datasets.


**Reasons To Reject:**

Substituting the causal masking layer in the G-Transformer results in a constrained level of innovation.

**Reproducibility:**

3: Could reproduce the results with some difficulty. The settings of parameters are underspecified or subjectively determined; the training/evaluation data are not widely available.

**Reviewer Confidence:**

3: Pretty sure, but there's a chance I missed something. Although I have a good feel for this area in general, I did not carefully check the paper's details, e.g., the math, experimental design, or novelty.

**Typos Grammar Style And Presentation Improvements:**

It is better to clarify the G-Transformer (AT) and G-Transformer (NAT) in the paper.

---

> ### Author Rebuttal · Authors · 2023-08-29
>
> Thank you for your constructive feedback.
>
> **Q1:** Substituting the causal masking layer in the G-Transformer results in a constrained level of innovation. \
> **A1:** We will revise the paper to clarify it. Substituting the causal masking layer in G-Transformer is only a necessary part at the attention level to apply sentence alignment, which relies model design and training loss to work. As Section 3.3 describes, we update the NAT models to ensure the sentence alignment modeling and training. For example, we introduce new length prediction per sentence in G-Trans+GLAT, improved CTC loss for sentence alignment in G-Trans+GLAT+CTC, and improved DAG design to restrict the transition in sentence in G-Trans+DAT. Without these new designs, the NAT models cannot be reliably trained.
>
> **Q2:** How does the model perform in terms of the model size? Can the larger model handle the longer input sequence issue? \
> **A2:** We will discuss it in the revised version. A larger model does not necessarily solve the longer input sequence issue given the same amount of training data. For pre-trained language models, a larger model typically shows stronger ability to handle long sequences because the increased parameters enable the model to capture more complex long-context dependencies. However, for a non-pretraining setting, a larger model does not necessarily show better performance on the current document-level MT benchmarks. The longer input makes the model more likely to overfit and a larger model will make it worse given the limited training corpus. Take the AT baseline G-Transformer as an example. When we increase the model size from Base to Big, the d-BLEU scores decline by 0.36, 1.41, and 0.10 on TED, News, and Europarl, respectively. When we further increase the model size to Large, the d-BLEU scores further decline by 16.53, 8.49, and 0.56 on TED, News, and Europarl, respectively.
>
> **Q3:** Why NAT+CTC and GLAT+CTC is worse than NAT and GLAT for the Europarl dataset in Table 1? \
> **A3:** We will revise the paper to discuss it. This abnormal phenomenon is observed for the first time, and we speculate it is caused by the CTC loss on long sequence. As Table 1 shows, both NAT/GLAT+CTC and NAT/GLAT encounter training failures on the raw Europarl dataset, and the failures are most likely caused by multi-modality and misalignment issues (Section 5.1). However, training with CTC loss on documents is not always stable, leading to occasional failures such as 0.00 d-BLEU scores of NAT/GLAT+CTC on the Europarl raw dataset. We speculate that the instability of CTC loss is caused by the increased alignment space between long input and output (Line 437).
>
> **Q4:** Typos and presentation. \
> **A4:** We will clarify the difference and the relation between the AT and NAT versions of G-Transformer. **G-Transformer (AT)** has an encoder-decoder architecture, involving two types of multi-head attention. One is for global document, naming global attention, while another is for local sentence, naming group attention. The global attention is simply a normal multi-head attention, which attends to the whole document. The group attention differentiates the sentences in a document by assigning a group tag to each sentence, which are used to calculate an attention mask to avoid cross-sentential attention. The two multi-head attentions are combined using a gate-sum module. G-Transformer (AT) uses group attention on low layers only but combined attention on top 2 layers. **G-Transformer (NAT)** follows the attention design, but removes the decoder casual mask for parallel decoding. Furthermore, the input and output of decoder are changed from autoregressive format to non-autoregressive, and NAT loss is used for training.

---

### Meta-Review · Area_Chair_RqYW · 2023-09-16

**Recommendation:** 3

**Metareview:**

This paper investigates the application of non-autoregressive transformers (NATs) to document-level MT. It proposes an alignment method for ensuring that generated text respects sentence boundaries, and compares NATs to standard autoregressive transformers, finding that the former produce poorer-quality document-level translations.

Reviewers felt that this was a substantial and novel contribution that establishes useful benchmark results for document-level NATs. They appreciated the comprehensive evaluation with different recent NAT architectures, and the honest comparison that demonstrates underperformance. Negatives included the use of small datasets for only one language pair, lack of statistical significance, and insufficient exploration of known techniques for speeding up the autoregressive baselines.

This is a solid contribution that establishes a useful new baseline technique and corresponding benchmark result for document-level work with NATs. It will serve as a good point of reference for people hoping to extend this line of work. The main downside is the somewhat narrow scope of the evaluation, but this is unlikely to seriously affect the conclusions.

I would like to highlight the use of BLEU rather than more accurate neural metrics such as COMET and BLEURT. Since this is typical of broad areas of MT research, it would be unfair to penalize this particular paper, but this is something that should change in the future, for high-resource settings.

---

### Decision · Program_Chairs · 2023-10-07

**Decision:**

Accept-Findings

**Comment:**

This paper investigates the application of non-autoregressive transformers (NATs) to document-level MT. It proposes an alignment method for ensuring that generated text respects sentence boundaries, and compares NATs to standard autoregressive transformers, finding that the former produce poorer-quality document-level translations.

Reviewers felt that this was a substantial and novel contribution that establishes useful benchmark results for document-level NATs. They appreciated the comprehensive evaluation with different recent NAT architectures, and the honest comparison that demonstrates underperformance. Negatives included the use of small datasets for only one language pair, lack of statistical significance, and insufficient exploration of known techniques for speeding up the autoregressive baselines.

This is a solid contribution that establishes a useful new baseline technique and corresponding benchmark result for document-level work with NATs. It will serve as a good point of reference for people hoping to extend this line of work. The main downside is the somewhat narrow scope of the evaluation, but this is unlikely to seriously affect the conclusions.

I would like to highlight the use of BLEU rather than more accurate neural metrics such as COMET and BLEURT. Since this is typical of broad areas of MT research, it would be unfair to penalize this particular paper, but this is something that should change in the future, for high-resource settings.